# Co-Loading of Black Phosphorus Nanoflakes and Doxorubicin in Lysolipid Temperature-Sensitive Liposomes for Combination Therapy in Prostate Cancer

**DOI:** 10.3390/ijms25010115

**Published:** 2023-12-21

**Authors:** Chandrima Das, Cristina Martín, Sebastian Habermann, Harriet Rose Walker, Javed Iqbal, Jacobo Elies, Huw Simon Jones, Giacomo Reina, Amalia Ruiz

**Affiliations:** 1Institute of Cancer Therapeutics, School of Pharmacy and Medical Sciences, Faculty of Life Sciences, University of Bradford, Bradford BD7 1DP, UK; c.daschandrima@gmail.com (C.D.); h.r.walker@bradford.ac.uk (H.R.W.); javediqbal.georgian@gmail.com (J.I.); j.eliesgomez@bradford.ac.uk (J.E.); h.s.jones@bradford.ac.uk (H.S.J.); 2Department of Bioengineering, Universidad Carlos III de Madrid, 28911 Leganés, Spain; cristima@ing.uc3m.es; 3Empa Swiss Federal Laboratories for Materials Science and Technology, Lerchenfeldstrasse 5, 9014 St. Gallen, Switzerland; sebastian.habermann@empa.ch

**Keywords:** black phosphorus nanoflakes, phototherapy, photodynamic therapy, LTSL

## Abstract

Black phosphorus (BP) is one of the most promising nanomaterials for cancer therapy. This 2D material is biocompatible and has strong photocatalytic activity, making it a powerful photosensitiser for combined NIR photothermal and photodynamic therapies. However, the fast degradation of BP in oxic conditions (including biological environments) still limits its use in cancer therapy. This work proposes a facile strategy to produce stable and highly concentrated BP suspensions using lysolipid temperature-sensitive liposomes (LTSLs). This approach also allows for co-encapsulating BP nanoflakes and doxorubicin, a potent chemotherapeutic drug. Finally, we demonstrate that our BP/doxorubicin formulation shows per se high antiproliferative action against an in vitro prostate cancer model and that the anticancer activity can be enhanced through NIR irradiance.

## 1. Introduction

Personalised, smart, and targeted therapies are the present frontiers of cancer treatment. In this context, novel approaches, including drug cocktails or combining chemo with radio/immunotherapies, are at the boundary of clinical advancement. At the same time, new techniques and materials are in the spotlight to induce better therapeutic outcomes to current radical therapies while reducing the associated side effects. In this context, phototherapies (PTs) have emerged as one of the leading therapeutic strategies [1]. PTs are, per se, personalised, smart, and targeted approaches, where the light dose can be easily tailored, and it is spatially confined and tailored by the irradiance fibre used from a few microns (size of single cells) up to centimetres (size of solid tumours). The advantage of a minimally invasive treatment where the ablation of organ-confined tumours eliminates cancerous tissue and avoids the side effects of a radical therapy has been evaluated in clinical trials for the treatment of prostate cancer (Lutex^®^ and TOOKAD^®^) [2]. However, photosensitisers tend to accumulate in the skin, where they can be activated by sunlight or artificial room light for several weeks after administration. Therefore patients receiving this therapy require protection from light until the drug has been completely cleared from the body [3]. It is a desirable system that can be activated in the tumour vasculature within minutes of light irradiance, and it is cleared rapidly. This rapid clearance will allow for the delivery of the drug/photosensitiser and light in the same session and avoids the need for prolonged light protection [3].

Advanced nanomaterials like graphene oxide or black phosphorus are pioneers in the biomedical field in providing multiple therapeutic functions in a single nanoplatform that allows for targeting, drug delivery, and photo- and immunotherapy [4]. A precise light-targeted treatment can be administered owing to these nanomaterials’ photo-absorbing capacity; however, this approach can be hindered by the limited depth of light penetration, resulting in potentially incomplete ablation and the relapse of the tumour [5]. Nanosized black phosphorus has recently gained attention owing to its photodegradation mechanism; it can be used directly as a photosensitiser, generating a high quantity of reactive oxygen species (ROS) together with the hyperthermia induced during near-infrared irradiance [6].

Delivering effective chemotherapeutic doses to the tumour whilst reducing systemic toxicity is a must to increase the selectivity of cancer treatments. Doxorubicin (Dox) is a potent chemotherapeutic drug offered as a course of several cycles of treatment in cancer therapy. However, Dox-induced cardiotoxicity results in the loss of functional myocytes and irreversible heart injury [7]. To circumvent this limitation, ThermoDox^®^ has been developed as a formulation containing low-temperature-sensitive liposomes (LTSLs), for localised, on-demand, and ultrafast Dox release in the tumour vasculature [8]. To date, in preclinical studies, researchers have triggered Dox release from LTSLs using water bath hyperthermia [9], high-focused ultrasound [10], or via the photothermal properties of magnetic nanoparticles embedded in the bilayer of LTSLs [11]. In the present work, we engineered for the first time a ThermoDox^®^ formulation that consists of DPPC, MSPC (lysolipid), and DSPE-PEG2000 (86:10:4 M ratio) [8], decorated with black phosphorus nanoflakes (BPNFs). LTSLs loaded with BPNFs and Dox were designed to integrate the photothermal and photodynamic capacity of BP with the chemotherapeutic effects of Dox in a light-triggered nanosystem for the efficient treatment of organ-confined tumours in prostate cancer. This multifunctional therapeutic nanoplatform has great potential in the treatment of men with organ-confined prostate cancer.

## 2. Results and Discussion

Before encapsulation, BPNFs were prepared using a liquid exfoliation method [12,13]. As depicted in Figure 1, the liquid exfoliation procedure involved grinding and extensive sonication of BP crystals in 1-methyl-2-pyrrolidone (NMP). The ultrasmall BPNFs obtained via centrifugation were dispersed in methanol and immediately coated with DOPE to avoid material degradation since BP is sensitive to water and oxygen. The entire process was carried out with the material protected from direct light exposure since BP can also be oxidised under visible-light irradiance.

TEM micrographs revealed the presence of primary flakes with a size of 12 ± 7 nm (Figure 1a,b). The hydrodynamic size of the flakes’ methanolic suspension was 154.2 (±3.31) nm, as measured with the DLS method, and the PDI value was 0.225 (±0.02) (Table 1). This size (Z-average < 200 nm) and the polydispersity of the sample (PDI < 0.3) are desirable for the intravenous administration of nanocarriers and indicate a homogenous population of the nanomaterial. These characterisation results are similar to the ones reported by Geng and coworkers [12]. TEM is considered the gold standard for imaging nanoparticles/nanomaterials, particularly those with high electron density. However, the preparation of the sample for TEM analysis and the vacuum environment of the microscope lead to the collapse of any aggregates or agglomerates that can exist in the colloidal suspension. The slight increase in the size of the BPNFs from TEM analysis to DLS characterisation is associated with the assembly of small agglomerates of BPNFs coated with DOPE. The amine group of the phospholipids’ heads is believed to interact with the surface of BPNFs during the exfoliation process through ionic interaction. However, when transferring the methanolic suspension of BPNFs to aqueous media to characterise its hydrodynamic size, the formation of small aggregates can occur due to the hydrophobic tales of the phospholipid repelling the water.

Figure 1c shows the average Raman spectra of bulk (top) and exfoliated (bottom) BP materials. The three characteristic Raman vibration modes, namely A^1^_g_, B_2g_, and A^2^_g_, of few-layer black phosphorus were recorded. Here, the A^1^_g_ mode corresponds to the out-of-plane phonon modes, with the phosphorus atoms in the top and bottom sublayers vibrating in opposite directions, whereas the B_2g_ and A^2^_g_ phonon modes correspond to the vibration along the in-plane zigzag and armchair directions with adjacent atoms vibrating out of phase, respectively [14]. The bulk material displayed the three peaks at 353.7 cm^−1^, 426.3 cm^−1^, and 453.5 cm^−1^, respectively. However, a shift in the spectrum to larger frequencies was visualised for the exfoliated nanomaterial, with active phonon modes A^1^_g_ at 362.2 cm^−1^, B_2g_ at 439 cm^−1^, and A^2^_g_ at 465.8 cm^−1^. This blue-shifting phenomenon points to the formation of BPNFs due to the efficient exfoliation process, as previously reported in the literature [15,16]. Furthermore, the change in the intensity ratio of A^2^_g_ and A^1^_g_ from the bulk black phosphorus crystal (1.2) to the BPNFs (1.8) demonstrates that BPNFs possess a few-layer structure [16,17]. Finally, the phosphate peak at ~960 cm^−1^ was not observed in any of the modes.

LTSLs (DPPC:MSPC:DSPE-PEG2000, at 86:10:4 M ratio) were prepared using the thin-lipid film hydration method. Since BPNFs were coated with DOPE, they could easily be incorporated into the lipid bilayer via hydrophobic interactions with the phospholipids of the lipid mixture. BPNF loaded in the liposomal vesicles are depicted in Appendix A. The lipid film was hydrated with a slightly acidic buffer of (NH_4_)_2_SO_4_ (pH 5.4). In this way, the internal core of the liposomes could be used to load Dox via pH gradient after exchanging the external buffer with HBS (pH 8.5). Then, pre-formed LTSL-BPNFs were incubated with Dox at a lipid–drug weight ratio of 20:1 at 37 °C for 90 min, followed by purification using size-exclusion chromatography. Large aggregates of unloaded BPNFs could be separated from the liposomal fractions using the PD-10 column (Appendix A). The colloidal properties of LTSLs loaded with BPNFs and Dox using the remote-loading method are summarised in Table 1. The hydrodynamic size of LTSLs increased slightly after loading them with BPNFs and Dox. The slight change in the surface charge observed between LTSL-BPNFs and LTSL-BPNF-Dox could be due to the changes in the buffers during the loading process. LTSL-BPNF liposomes were hydrated using 240 mM (NH_4_)_2_SO_4_. After Dox loading, the liposomes were purified in a PD-10 column using an HBS, 20 mM of HEPES, and 0.8 wt% NaCl, at pH 7.4. The same particle prepared in a buffer with different salts or different concentrations will have changes in the absolute zeta potential value. Nevertheless, co-loaded liposomes exhibited small hydrodynamic sizes (135.8 nm (±2.01)), PDI below 0.3 (0.277 (±0.01)), and slightly negative Z-potential (−11.4 mV (±1.82)).

ThermoDox^®^ is the most advanced LTSL formulation, exhibiting a burst release profile in the presence of mild hyperthermia (41–42 °C). The optimal loading of Dox into LTSLs has been achieved through transmembrane pH-gradient methods to drive Dox loading inside the liposomes. However, it has often been reported that the Dox loading capacity in the LTSL formulation (ThermoDox^®^) is relatively low [18,19,20]. Currently, there are no reports assessing the effect of black phosphorus quantum dots on the encapsulation efficiency of Dox into LTSLs. With an optimum lipid–Dox weight ratio of 20:1 and a lipid–BPNF weight ratio of 10:1, we obtained 82.7% (±12.2) and 57.4% (±0.5) for Dox and BPNF EE%, respectively, with n = 3 independent replicas of the formulation. The mechanism of remote loading via (NH_4_)_2_SO_4_ is based on the low solubility of Dox–sulphate crystals in the core of the liposomes and our LTSL-BPNF-Dox EE% agrees with our previous studies using this loading method [9]. In summary, the physicochemical characterisation of the liposomes indicates that the presence of BPNFs did not adversely affect the colloidal properties of the formulation. Furthermore, based on our results, LTSL-BPNF-Dox exhibited high Dox-loading efficiency (>80%), suggesting that the presence of the BPNFs in the lipid bilayer did not affect Dox loading.

The rapid oxidation of BPNFs in PBS has been previously reported, where the absorbance of the material in PBS decreased significantly with time and the solution became colourless after 72 h [12]. We decided to investigate the long-term stability of LTSL-BPNFs in HBS buffer, at pH 7.4, mimicking physiological conditions (Figure 2). As observed in Figure 2a, the absorbance LTSL-BPNFs exhibits a broad absorption band spanning the ultraviolet and visible regions of the spectrum [21], whilst LTSL absorbance is detected only in the UV region around 260 nm. The broad peak with the maximum at 500 nm observed for LTSL-BPNF-Dox corresponds with the characteristic peak of doxorubicin. Figure 2b,c show the variation in the absorbance of the formulation at 450 nm up to 13 days stored at room temperature or 4 °C. In both cases, the samples were protected from light to reduce photo-oxidation, and the ratio between the intensity at different time points was compared with the initial absorbance value. The sample stored at 23 °C decreased by 40% after 13 days, whilst the sample kept in the fridge at 4 °C was more stable and only reduced its intensity by 17%. The degradation of black phosphorus in water is due to the irreversible oxidation of P to phosphate groups. The initial capping reaction with DOPE and embedding the BPNFs in the bilayer of LTSLs greatly increased the stability of the nanomaterial.

The photoactivatable release of the drug was evaluated in a test tube and PC-3-cell-cultured monolayers (Figure 3). For test tube experiments, we decided to investigate the release of the drug at 23 (room temperature) and 37 °C (physiological temperature) (Figure 3a). A five-fold increase in drug release was observed when the sample was incubated at 37 °C, compared with RT after 20 min of laser irradiance at 1W/cm^2^ power density. Lysolipid-containing thermosensitive liposomes (LTSLs) enable the burst release of Dox upon phase transition because the permeability of the membrane is increased at the liquid crystalline phase under mild hyperthermia (40–42 °C) [22]. By starting the incubation at 37 °C, the amount of BPNFs loaded in the bilayer of LTSLs can induce a local temperature increase in the bilayer of LTSLs, reaching mild hyperthermic conditions. These local temperature changes can lead to the stabilisation of transient membrane pores in the bilayer, allowing for transitions between gel–liquid and crystalline phases and inducing Dox release [23]. Al-Jamal et al. described the pharmacokinetic profile of Dox-loaded LTSLs in B16F10 tumour-bearing mice (C57BL6) in the presence/absence of mild hyperthermia. The highest Dox accumulation observed for this thermosensitive formulation in vivo is presumably due to the rapid intravascular release of the drug in the heated tumour in the first 25 min after the injection [24].

To study the in vitro release of LTSL-BPNF-Dox, a release/uptake study was performed using the MuviCyte Live-Cell Imaging Kit (Figure 3b). For this purpose, PC-3 cells were incubated with the formulation for 1h, and then the cells were irradiated at 1 W/cm^2^ for 10 min and returned to the incubator to allow for the uptake of the Dox released during laser-induced hyperthermia. Since the fluorescence of Dox is not quenched after its release, its subcellular localisation in PC-3 cells can be tracked using the RFP channel of the MuviCyte [9,25]. As a control for the evaluation of non-specific leakage of the formulation, the cells were incubated with LTSL-BPNF-Dox, but they were kept in the dark at 37 °C over the same period. Negligible Dox signals were detected in the cells incubated in the dark, suggesting the good stability of the formulation even in the presence of 10% serum in cell culture media. As depicted in the micrographs of the cells irradiated with the NIR laser, the cells showed distinctive signals of intracellular Dox accumulation. However, Dox signals are lower in intensity than the ones reported in a similar release/uptake study carried out by Geng et al., due to the difference in Dox concentration between our studies [26]. These authors carried out their in vitro release experiment by incubating MCF-7 cells with BP dot–Dox-loaded lipid nanocapsules at 20 µg/mL of Dox (34.5 µM), whilst we performed our experiment at 2 µM. Our choice of dose was based on Dox IC_50_ for PC-3 cells (IC_50_ 1.08 µM) so that we could assess a more clinically relevant scenario [27]. Moreover, we have previously described how LTSLs can be loaded with Dox at different drug–lipid weight ratios using the ammonium sulphate pH gradient without affecting their burst release profile under hyperthermia. This flexibility is an excellent advantage for translating the formulation from the bench to the clinic. Overall, light-activated Dox release was obtained in vitro using LTSL-BPNF-Dox compared to non-irradiated samples.

To further investigate the potential of LTSL-BPNF-Dox for combination therapy (photodynamic + chemotherapy) in prostate cancer, the ROS generation capacity was evaluated using the permeable probe DHE, which allows for the detection of superoxide radicals [28]. As exhibited in Figure 4, the increased fluorescence intensity of the DHE after laser irradiance suggests a superoxide generation capability during the therapeutic treatment. These results are encouraging since the generation of species like superoxide or peroxide is not limited by the oxygen levels during a photodynamic treatment [29]. Therefore, the use of a modified formulation of ThermoDox^®^ using LTSL-BPNF-Dox, activated in the tumour microenvironment, where hypoxia plays a significant role, will improve the therapeutic outcome of the therapy.

Having confirmed the capacity of LTSL-BPNF-Dox to generate ROS after NIR laser irradiance, we investigated the potential of the liposomes for combinatory phototherapy + chemotherapy. An activatable formulation like ThermoDox^®^ enables the intravascular Dox release in heated tumours [18]. A successful nanosystem should maintain the thermosensitive characteristic of LTSLs, which can be activated in the tumour vasculature within minutes of light irradiance, together with an improved photodynamic capacity due to the presence of BPNFs in the structure. Prostate cancer cells, PC-3 and DU 145, were treated for 1 h with the LTSL-BPNF-Dox liposome, at 0.25 or 0.5 mM liposome concentration (1 or 2 μM Dox respectively). After 1h, the cells were irradiated with the laser at 1 W/cm^2^ power density for 10 min to induce the release of Dox. After 3 h, the cells were washed, and toxicity was assessed after 24 h post-treatment using a resazurin assay. Liver Hep G2 cells were included as a control since this organ is the most likely to encounter the nanoparticles after intravenous administration. As observed in Figure 5a–c, the incubation of the cells with LTSL-BPNFs up to 0.5 mM liposome concentration for 3 h after laser treatment did not cause a significant change in cell viability. A slight drop (but not significant *p* > 0.05) in cell viability to ~75–80% was observed for the prostate cancer cell lines, which could be attributed to some photodynamic effects associated with the presence of the BPNFs loaded in the liposomes. However, according to the EN ISO 10993-5 guideline, significant cytotoxic effects will be considered below 70% [30].

PC-3 and DU 145 cells treated with LTSL-BPNF-Dox (Figure 5d,e) that were irradiated revealed a significant reduction in cell viability (~55–65%, 0.25 mM, ** *p* < 0.01 or ~45%; 0.5 mM, *** *p* < 0.001) compared with non-irradiated cells. Previous results of our group have demonstrated the stability of LTSLs loaded with Dox in cell culture media in the absence of hyperthermia [9]. In that study, PC-3 and CT-26 cells did not show a reduction in cell viability in similar experimental conditions, so we can rule out the possibility of non-specific leakage from LTSLs in vitro. This was further confirmed by our thermosensitivity studies in vitro (Figure 4), in which only negligible Dox signals were detected in non-irradiated cells. The reduction in cell viability to ~75% for non-irradiated cells could be due to the degradation of BPNFs in physiological conditions. Banno et al. described the loss of lysolipids in the bilayer within minutes of contact with serum proteins in biological media [31]. The desorption of lysolipids could induce structural changes in the bilayer, which exposes BPNFs to the media on the surface of LTSLs or releases them; therefore, they rapidly degrade if they are not protected by the lipid bilayer. Before completely degrading to harmless phosphate ions, BPNFs are oxidated by the solvent and by molecular oxygens with a still unknown reaction mechanism. The oxidation of BP flakes produces a pool of oxygenated radical species through a mechanism that is still not clear. Different studies demonstrated that superoxide ions are formed, suggesting the electron transfer between the electron-rich BP surface to the O_2_ molecules [32,33]. On the other hand, ^1^O_2_ can be generated through irradiance on BP nanoflakes or in composites doped with gold making it a versatile photodynamic agent [6,34]. In our case, BPNFs were shielded from O_2_ molecules by the lipid bilayer, hampering the BPNF degradation, so their radical formation induced toxicity. NIR irradiance induces the breaking of the liposomes, facilitating BPNF oxidation, which promotes ROS formation and can be directly associated with higher toxicity for cancer cells. It is worth mentioning that the final products of the degradation of BPNFs are phosphate ions, which are harmless by-products that will enter the natural metabolic pathway in the organism after BPNFs fulfil their therapeutic role [33].

## 3. Materials and Methods

### 3.1. Materials

Black Phosphorus (bulk) was purchased from Ossila Ltd., Sheffield, UK (Product Code M2106C1). All chemicals were purchased from Sigma-Aldrich, Gillingham, UK. 1,2-Dipalmitoyl-sn-glycero-3-phosphocholine, (Lipoid PC 16:0/16:0 (DPPC), 1,2-Dioleoyl-sn-glycero-3-phosphoethanolamine, (DOPE Lipoid 565600), and [N-(Carbonyl-methoxy polyethylene glycol-2000)-1,2-distearoyl-sn-glycero-3-phosphoethano[amine, sodium salt]] (DSPE-PEG2000 Lipoid 588200) were generous gifts from Lipoid GmbH (Ludwigshafen, Germany). 1,2-Dipalmitoyl-sn-glycero-3-phosphocholine and 1-stearoyl-2-hydroxy-sn-glycero-3-phosphocholine (MSPC) were purchased from Avanti Polar Lipid (Alabaster, AL, USA). Doxorubicin hydrochloride (Dox·HCl) was obtained from Biosynth Ltd., Compton, UK (Product Code: AD15377). Roswell Park Memorial Institute (RPMI) 1640 Medium, Dulbecco’s PBS (1x), penicillin–streptomycin solution liquid (10,000 units/mL), glutamine supplement (200 mM), and 0.25% trypsin/EDTA were purchased from ThermoFisher Scientific, Winsford, UK. Heat-inactivated foetal bovine serum (FBS) (11580516) was obtained from Gibco Life Technologies, Paisley, UK.

### 3.2. Methods

#### 3.2.1. Synthesis of BPNFs

The BPNFs were synthesised via a liquid exfoliation method [12]. Briefly, 100 mg of BP powder was added to 1 mL of 1-methyl-2-pyrrolidone (NMP), and the material was finely ground using a pestle and mortar, followed by 12 h sonication in a water bath sonicator at (800 W), reaching a final volume of 10 mL of NMP. The temperature was maintained below 35 °C at all times using a water reflux system. The resulting dispersion was centrifuged at 5000 rpm for 20 min, and the supernatant containing BPNFs was decanted gently to separate from the large aggregates. To purify the BPNFs and remove the NMP, the supernatant containing the BPNFs was mixed in a 1:1 *v*:*v* ratio with methanol and centrifuged at 13,000 rpm for 2 h. The resulting precipitate was mixed with DOPE at a 1:5 BPNF:DOPE weight ratio and dissolved in methanol. The mixture was sonicated for 30 min and left stirring for an extra hour to allow the full coating of the BPNFs to develop. Next, the sample was dried by flushing with N_2_ and kept at 4 °C protected from light.

#### 3.2.2. Preparation of LTSL-BPNFs

Phospholipids and BPNFs (10:1 weight ratio) were dissolved in a mixture of chloroform and methanol (4:1 *v*/*v*) and transferred to a 25 mL round-bottom plate. Liposomes were prepared using the lipid film hydration method. Briefly, the organic solvents were evaporated under reduced pressure at 55 °C for 1 h using a rotary evaporator (BÜCHI, Labortechnik AG, Flawil, Switzerland) and then flushed with an N_2_ stream to remove any residual traces of organic solvent. To achieve a final lipid concentration of 5 mM, the dried lipid films of LTSLs (DPPC:MSPC:DSPE-PEG2000 at 86:10:4 molar ratio) containing BPNFs were hydrated for 20 min with ammonium sulphate pH 5.4 (240 mM (NH_4_)_2_SO_4_). Following hydration at 60 °C, liposomes were downsized via sonication using a probe sonicator (Sonics VC505, Fisher Scientific, Loughborough, UK) through 3 cycles of 30 s ON and 10 s OFF, with 40–41% amplitude. After sonication, liposomes (LTSL-BPNFs) were allowed to anneal for a minimum of 2 h at RT.

#### 3.2.3. Dox Loading into LTSL-BPNFs

(NH_4_)_2_SO_4_ buffer was exchanged in the LTSL-BPNF formulation with HEPES buffer saline (HBS, 20 mM of HEPES and 0.8 wt% NaCl, pH 8.5) using a Sephadex™ G-25 PD-10 column (Cytiva, Fisher, UK) to generate the pH gradient required to load doxorubicin. After buffer exchange, LTSL-BPNFs were incubated with Dox·HCl at a lipid–Dox ratio of 20:1 weight ratio at 37 °C for 90 min [9]. Following incubation, unencapsulated Dox was removed using a PD-10 column equilibrated with an HBS, 20 mM of HEPES, and 0.8 wt% NaCl, at pH 7.4. Dox encapsulation efficiency (EE) was determined by measuring fluorescence intensity at λ_ex_ = 485 nm and λ_em_ = 590 nm using a Flexstation III plate reader. Dox-loaded LTSL-BPNFs before and after purification were lysed with DMSO (1:10 dilution), and Dox EE% was calculated using Equation (1):(1)$$EE\%=\frac{Drug\;concentration\;(after\;purification)}{Drug\;concentration\;(before\;purification)}\times 100$$

#### 3.2.4. Characterisation of BPNFs

The structural characterisation of BPNFs after exfoliation was carried out using Raman spectroscopy. Raman spectra of dry samples were acquired using a Renishaw inVia microscope equipped with a 514 nm laser. A 100× objective lens was employed, and the laser power and the exposure time were 1% and 3 s, respectively, in all the experiments. Different areas of the sample were tested, and at least 10 different spectra were obtained to represent the average spectra shown in this paper. UV–Vis absorption spectra were recorded on a Cary 100 (Varian) spectrophotometer at room temperature using a slit width of 0.4 nm and a scan rate of 600 nm/min. Transmission electron microscopy was performed with a Zeiss EM 900 microscope (Carl Zeiss Microscopy GmbH, Jena, Germany) at 80 kV.

#### 3.2.5. Colloidal Properties

Hydrodynamic size and ζ-potential (ZP) measurements were performed using a Zetasizer NanoZS90 (Malvern Panalytical Ltd., Malvern, UK). Size and ZP measurements were performed using disposable polystyrene cells and plain folded capillary zeta cells (Malvern Panalytical Ltd., Malvern, UK), respectively. BPNF suspensions in methanol or aqueous liposome samples were diluted 10-fold in deionised water for size measurements. Liposomes were diluted 100-fold for ZP measurements. All measurements were performed at 25 °C. Size measurements were performed with 3 measurements, each with 15 scans, while ZP measurements were performed with 6 measurements, each with 20 scans.

#### 3.2.6. Near-Infrared (808 nm) Laser-Induced Dox Release

This procedure was carried out using 48-well plates containing 400 µL of 0.5 mM LTSL-BPNF-Dox liposomes. The temperature of the liposomes was maintained at 23 or 37 °C using a water bath. The samples were irradiated with an 808 nm near-infrared (NIR) laser with a power density of 1 W/cm^2^ (LDM90, L808P1000MM laser diode set, Thorlabs, Cambridgeshire, UK). The 48-well plate was placed at a 10 cm distance from the laser diode and aligned so that the laser beam covered the entire surface of the well to ensure the uniform irradiance of the sample. Briefly, 50 μL of the samples were taken at each time point and further diluted in HBS, pH 7.4, to a final volume of 200 μL. Samples were then transferred into a 96-well black, clear flat-bottom plate, and fluorescence intensity was quantified at λ_ex_ = 485 nm and λ_em_ = 590 nm using a Flexstation III plate reader. The percentage of Dox released at each time point was calculated using Equation (2):(2)$$Dox\;release\;(\%)=\frac{\left(I(s)-I(0)\right)}{I(t)-I(0)}\times 100$$
where *I*(*s*) is the fluorescence intensity of samples at various time points, *I*(0) is the fluorescence intensity of Dox-loaded liposomes (background), and *I*(*t*) is the fluorescence intensity of the liposome suspension at time *t* = 0 h after lysis with DMSO (1:10 dilution).

#### 3.2.7. Cell Culture

Human prostate adenocarcinoma derived from the bone metastatic site (PC-3) (CRL-1435), human prostate carcinoma derived from the brain (DU 145) (HTB-81), and a liver cell line exhibiting epithelial-like morphology that was isolated from hepatocellular carcinoma (Hep G2) (HB-8065) were obtained from American Type Culture Collection (ATCC, Manassas, USA). Cells were cultured in RPMI 1640 (ThermoFisher Scientific, Winsford, UK), supplemented with 10% heat-inactivated FBS, 50 U/mL penicillin, 50 μg/mL streptomycin, and 1% L-glutamine, and maintained in an incubator at 37 °C and 5% CO_2_.

#### 3.2.8. Dox Uptake Study

PC-3 cells were seeded overnight in 12-well plates (1 × 10^5^ cells/well, 1 mL/well). The next day, cells were incubated for 1 h with LTSL-BPNF-Dox at 0.5 mM liposome concentration. The plates were sealed with parafilm, and the laser beam was focused on the treated well (other wells were covered to avoid crossed laser treatment) at a fixed distance of 10 cm with the desired power density. Cells were then irradiated for 10 min at 1 W/cm^2^ laser power density. After irradiance, the cells were put back into the incubator at 37 °C and 5% CO_2_ for another 3 h. Prior imaging, cells were washed to remove particles and excess free drug from the supernatant; then, the nuclei of the cells were counterstained with 1 μg/mL of Hoechst 33342 (ThermoFisher, Winsford, UK) for 10 min. After washing with PBS (x3), live images were recorded using a MuviCyte Live-Cell Imaging Kit (Perkin Elmer, Beaconsfield, UK).

#### 3.2.9. ROS Production

Intracellular reactive oxygen species (ROS) production was assessed using Dihydroethidium (ThermoFisher Scientific, Winsford, UK). First, cells were seeded overnight in 96-well plates (2 × 10^4^/well, n = 4 per condition). The next day, cells were incubated with 0.5 mM LTSL-BPNFs, followed by laser irradiance (1 W/cm^2^ power density, 10 min). The treatment was washed, and cells were incubated with the fluorescent probe dihydroethidium (DHE) which is a superoxide indicator. DHE was dissolved in Hanks’ balanced salt solution (HBSS) phenol red-free at 1 µM, and cells were incubated for 30 min. After that, the cells were washed twice with HBSS, and they were returned to the incubator; images were then recorded using a MuviCyte Live-Cell Imaging Kit (Perkin Elmer, Beaconsfield, UK). This probe exhibits blue fluorescence in the cytosol until oxidisation, when it intercalates within the cell’s DNA, staining the nuclei with a bright red fluorescent when superoxide is detected.

#### 3.2.10. In Vitro Cancer Therapy

PC-3, DU 145, and Hep G2 cells (2 × 10^4^/well) were seeded overnight in a 96-well plate (2 × 10^4^/well, n = 4 per condition). Cells were incubated with 200 µL of 0.25 or 0.5 mM LTSL-BPNF-Dox. For the group treated with an 808 nm laser (1 W/cm^2^ power density, 10 min), the plate was sealed with parafilm, and the laser beam was focused on the treated well (other wells were covered to avoid crossed laser treatment) at a fixed distance of 10 cm with the desired power density. Then, the plates were unsealed and put back into a humidified incubator at 37 °C and 5% CO_2_ for another 3 h. Then, the cells were washed and replenished with fresh complete media and returned to the incubator for another 20 h. Cell viability was assessed using resazurin assay. Briefly, the cells were incubated with 0.01 mg/mL resazurin solution for 4 h. After incubation, the fluorescence of the cell culture supernatant was measured (λ_ex_ = 544 nm, λ_em_ = 590 nm) using a Flexstation III plate reader (Molecular Devices (UK) Limited, Berkshire, UK). Four replicates per condition were used. The results were expressed as the percentage of cell viability (mean ± SD) and normalised to control untreated cells.

## 4. Conclusions

In this study, we presented an innovative approach aimed at addressing the challenge involving the development of stable and concentrated BP suspensions through the utilisation of temperature-sensitive liposomes operating at low temperatures (42 °C). Furthermore, this technique enables the co-encapsulation of BP nanoflakes alongside doxorubicin. Notably, our findings underline that the BP/doxorubicin composite independently exhibits significant antiproliferative effects on PC-3 and DU 145 in vitro prostate cancer models. Moreover, the therapeutic potential can be amplified through NIR irradiance, thereby bolstering the anticancer activity. In conclusion, our research introduces a novel strategy to enhance the stability and therapeutic potential of BP-based formulations, showcasing a multifaceted approach that includes photothermal, photodynamic, and chemotherapeutic modalities to effectively combat cancer.

## Data Availability

Data are contained within the article and Appendix A.

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
