# Peer review of "Co-Loading of Black Phosphorus Nanoflakes and Doxorubicin in Lysolipid Temperature-Sensitive Liposomes for Combination Therapy in Prostate Cancer"

_ijms, 2023, doi:10.3390/ijms25010115_

Round 1
Reviewer 1 Report
Comments and Suggestions for Authors
This work is based on the development of liposomes based on black phosphorus nanoflakes and doxorubicin for prostate cancer treatment. Although the authors performed several experiments, it seems that the characterization of LTSL-BPNF was not sufficient from aspects of physicochemical property and biological activity. Therefore, it is thought that this manuscript needs major revision to be accepted for publication. The details are as follows.
1. The authors confirmed the formation of BPNF after exfoliation process of bulk BP by Raman spectra. However, TEM image of BPNF is not clear in ESI Figure 1 and it is required to obtain its clearer image. What is the height of this BPNF? In addition, the authors need to show TEM images of LTSL-BPNF.
2. What is the reason that Z-potential of LTSL-BPNF showed more negative value after Dox loading?
3. Have the authors measured the hydrodynamic size of LTSL-BPNF after 13 days of incubation?
4. The authors need to examine the Dox release profile from LTSL-BPNF-Dox for a more extended period (at least until the majority of Dox is released).
5. Although it seems that the somewhat high fluorescence of DHE was observed in LTSL-BPNF after irradiation (Figure 4), the authors need to examine the ROS generation capacity quantitatively.
6. In this work, the authors used 0.25~0.5 mM of LTSL-BPNF-Dox, which are regarded as very high concentrations. It would significantly reduce the practical applicability of this system. What is its cytotoxicity in lower concentrations? In addition, what is the cytotoxicity of LTSL-BPNF itself? The authors also need to examine its cytotoxicity in other cell lines (including other tissue cell lines for confirmation of its safety and other prostate cancer cell lines for confirmation of its effect).
7. In Figure 5, LTSL-BPQD à LTSL-BPNF
Author Response
This work is based on the development of liposomes based on black phosphorus nanoflakes and doxorubicin for prostate cancer treatment. Although the authors performed several experiments, it seems that the characterization of LTSL-BPNF was not sufficient from aspects of physicochemical property and biological activity. Therefore, it is thought that this manuscript needs major revision to be accepted for publication. The details are as follows.
We thank Reviewer 1 for their suggestions, we have prepared a revised version where we addressed all the points. We believe that the work is mature enough for publication.
- The authors confirmed the formation of BPNF after exfoliation process of bulk BP by Raman spectra. However, TEM image of BPNF is not clear in ESI Figure 1 and it is required to obtain its clearer image. What is the height of this BPNF? In addition, the authors need to show TEM images of LTSL-BPNF.
Our Response: We thank the referee for the highlight of the TEM analyses.
Our action: We included in Figure 1 the TEM characterisation of BPFN after exfoliation and the size analysis of the nanoflakes. The TEM image of LTSL-BPNF is provided in ESI Figure 1, and the nanoflakes loaded in the liposomes have been highlighted with arrows. The caption has been corrected for further clarification.
- What is the reason that Z-potential of LTSL-BPNF showed more negative value after Dox loading?
Our response: We thank the referee for highlighting this detail of the experiment. Dox loading was carried out via a remote loading method, which relies on generating a transmembrane pH gradient using (NH4)2SO4 to drive efficient DOX loading inside the liposome. The drug should be encapsulated in the core of the liposomes, thus not affecting the particle's surface charge. The slight change in the surface charge observed between LTSL-BPNF and LTSL-BPNF-Dox could be due to the changes in the buffers during the loading process. LTSL-BPNF liposomes are hydrated using 240 mM (NH4)2SO4. After Dox loading, the liposomes are purified in a PD-10 column using an HBS, 20 mM HEPES and 0.8 wt% NaCl, pH 7.4. The same particle prepared in a buffer with different salts or different concentrations will have changes in the absolute zeta potential value. This explanation has been added to the manuscript.
Our Action: We have included in the manuscript (page 8) the following text: “The slight change in the surface charge observed between LTSL-BPNF and LTSL-BPNF-Dox could be due to the changes in the buffers during the loading process. LTSL-BPNF liposomes are hydrated using 240 mM (NH4)2SO4. After Dox loading the liposomes are purified in a PD-10 column using a HBS, 20 mM HEPES and 0.8 wt% NaCl, pH 7.4. The same particle prepared in a buffer with different salts or different concentrations will have changes in the absolute zeta potential value.”
- Have the authors measured the hydrodynamic size of LTSL-BPNF after 13 days of incubation?
Our response: Due to the time frame provided to carry out the experiments, we couldn’t assess a long-term stability study. If the referee and the editor feel strongly about it, we will include the values in the manuscript later in the editing process.
- The authors need to examine the Dox release profile from LTSL-BPNF-Dox for a more extended period (at least until the majority of Dox is released).
Our response: We have chosen to study the release profile of LTSL loaded with BPNF and Dox in a window frame of 20 minutes since it has been described by Al-Jamal (2012, Biomaterials 33, 18, 4608-4617) that the circulation time of Dox-loaded LTSL in B16F10 tumour-bearing mice (C57BL6) decreased below 10% in presence or absence of mild hyperthermia. The highest Dox accumulation observed for this thermosensitive formulation in vivo is presumably due to the rapid intravascular release of the drug in the heated tumour, not to a prolonged circulation time like what can be achieved with Doxil-type formulations.
Our action: We have included a sentence in the manuscript highlighting this aspect (page 10): “Al-Jamal et al. has described the pharmacokinetic profile of Dox loaded LTSL in B16F10 tumour-bearing mice (C57BL6) in presence/absence of mild hyperthermia. The highest Dox accumulation observed for this thermosensitive formulation in vivo is presumably due to the rapid intravascular release of the drug in the heated tumour in the first 25 minutes after the injection.”
- Although it seems that the somewhat high fluorescence of DHE was observed in LTSL-BPNF after irradiation (Figure 4), the authors need to examine the ROS generation capacity quantitatively.
Our response: The data in Figure 4 shows an increase in DHE fluorescence (as a marker of ROS production) based on fluorescence microscopy experiments. This measurement method very much lends itself to a qualitative assessment of ROS generation (i.e., has it increased or decreased), as opposed to quantitative analysis (how much has it changed by?). Absolute quantification of ROS generation is challenging to do (especially with dyes such as DHE, which react with a number of different reactive oxygen species), so often relative quantification of ROS levels is used instead. Indeed, relying on fluorescence to detect DHE can be misleading because these probes generate ethidium, a non-specific oxidation product, and the O2•−-specific product 2-hydroxy ethidium. The challenge arises from the overlapping fluorescence spectra of these two products, making it difficult to distinguish the contribution of non-specific oxidation from O2•−-dependent oxidation (if any) to the overall fluorescence signal. [Murphy, Michael P., et al. "Guidelines for measuring reactive oxygen species and oxidative damage in cells and in vivo." Nature metabolism 4.6 (2022): 651-662.] For those reasons, we are unconvinced that expressing the visible change in fluorescence signal as a percentage or relative change adds value or changes the manuscript's narrative, so we have not included this in the revisions.
- In this work, the authors used 0.25~0.5 mM of LTSL-BPNF-Dox, which are regarded as very high concentrations. It would significantly reduce the practical applicability of this system. What is its cytotoxicity in lower concentrations? In addition, what is the cytotoxicity of LTSL-BPNF itself? The authors also need to examine its cytotoxicity in other cell lines (including other tissue cell lines for confirmation of its safety and other prostate cancer cell lines for confirmation of its effect).
Our response: The experiments were designed using a liposome concentration suitable to visualise the cell uptake of the doxorubicin using a live cell imager. Since our total incubation time is only 3 hours after laser irradiance, we need to incubate the cells with a relatively high concentration of liposomes (phospholipid concentration) to detect changes in fluorescence and cell viability. We have previously described how LTSL can be loaded with Dox at different drug:lipid weight ratio using the ammonium sulphate pH gradient without affecting their burst release profile under hyperthermia (J Control Release 2020 Dec 10:328:665-678). This flexibility is an excellent advantage for translating the formulation from the bench to the clinic.
Our action: We have assessed the cytotoxicity of LTSL-BPNF and LTSL-BPNF-Dox in presence-absence of laser irradiance using other prostate (DU 145) and non-prostate cell lines (Figure 5). HepG2 was selected because the liver is the organ that will encounter the majority of the nanoparticles after intravenous administration. However, with laser-induced hyperthermia in the tumour, the LTSL-BPNF-Dox formulation will have a burst release of the drug in the intravascular region of the tumour, maximising the accumulation of the drug there and sparing healthy non-irradiated tissues.
- In Figure 5, LTSL-BPQD à LTSL-BPNF
Our action: The typo in Figure 5 has been corrected.

Reviewer 2 Report
Comments and Suggestions for Authors
The aim of this research article is to link the innovative potential of nanotechnology with cancer therapies. Primarily, it focuses on the development of a novel approach to stabilise black phosphorus nanoflakes (BPNF) in temperature-sensitive lysolipid liposomes, a strategy that aims to mitigate the rapid degradation of BPNF in the presence of oxygen and thus remove a major obstacle in cancer treatment. The objectives of the study extend to the co-encapsulation of BPNF with doxorubicin, an established chemotherapeutic agent, in these liposomes to exploit synergistic effects. The study will also investigate the efficacy of this combined formulation in in vitro trials on a prostate cancer model. A critical element of the research is to determine whether irradiation with near-infrared radiation could potentially enhance the anti-cancer effects of the liposomal BPNF/doxorubicin formulation.
The study comprises a methodological investigation of liposomal encapsulation techniques to improve the bioavailability and stability of BPNF for medicinal purposes. It covers the formulation process aiming at high encapsulation efficiency and stability in biological environment. The paper describes the investigation pathway involving the interaction of this novel composite with near-infrared radiation to utilise the photothermal properties of BPNF for improved therapeutic outcomes in cancer therapy. The framework thus includes a detailed in vitro analysis in a controlled prostate cancer model to gain insights that could lead to further in vivo and clinical studies.
The manuscript presents original results and makes a valid contribution to the field. However, there are a few issues that should be addressed before can be considered for publication:
1) The actual laser irradiance used is not clear. The value given (1 W/cm^2) is an enormous value, orders of magnitude higher than that normally used. The author should state how this value was determined and whether it is the nominal value of the laser or the value of the sample.
2) In lines 347, 365, 378, 413, 132, the author uses the term “laser power" and the value of 1 W/cm^2. This is not correct. It is a power density and when referring to electromagnetic radiation the term irradiance should be used (see IUPAC Recommendations 2006). The authors should correct this.
Photothermal and photodynamic therapies are carried out with very different irradiance levels. In photodynamic, the irradiance should be low to avoid heating effects. For photothermal effects, it is common to use higher irradiance levels to heat the area to be treated. The authors refer to the use of photothermal effects, but do not show that they lead directly to heating. This should be clarified by the authors.
Author Response
The aim of this research article is to link the innovative potential of nanotechnology with cancer therapies. Primarily, it focuses on the development of a novel approach to stabilise black phosphorus nanoflakes (BPNF) in temperature-sensitive lysolipid liposomes, a strategy that aims to mitigate the rapid degradation of BPNF in the presence of oxygen and thus remove a major obstacle in cancer treatment. The objectives of the study extend to the co-encapsulation of BPNF with doxorubicin, an established chemotherapeutic agent, in these liposomes to exploit synergistic effects. The study will also investigate the efficacy of this combined formulation in in vitro trials on a prostate cancer model. A critical element of the research is to determine whether irradiation with near-infrared radiation could potentially enhance the anti-cancer effects of the liposomal BPNF/doxorubicin formulation.
The study comprises a methodological investigation of liposomal encapsulation techniques to improve the bioavailability and stability of BPNF for medicinal purposes. It covers the formulation process aiming at high encapsulation efficiency and stability in biological environment. The paper describes the investigation pathway involving the interaction of this novel composite with near-infrared radiation to utilise the photothermal properties of BPNF for improved therapeutic outcomes in cancer therapy. The framework thus includes a detailed in vitro analysis in a controlled prostate cancer model to gain insights that could lead to further in vivo and clinical studies.
The manuscript presents original results and makes a valid contribution to the field. However, there are a few issues that should be addressed before can be considered for publication:
We express our gratitude to Reviewer 2 for their valuable suggestions. In response, we have incorporated all the suggested revisions in the updated version. We are confident that the work has reached a level of maturity suitable for publication
1) The actual laser irradiance used is not clear. The value given (1 W/cm^2) is an enormous value, orders of magnitude higher than that normally used. The author should state how this value was determined and whether it is the nominal value of the laser or the value of the sample.
Our response: We are sorry about this. We used the nominal laser irradiance. The power density value is in the typical range for PT cancer treatment. " Bastiancich, C., Da Silva, A., & Estève, M. A. (2021). Photothermal therapy for the treatment of glioblastoma: potential and preclinical challenges. Frontiers in Oncology, 10, 610356. Qin, J., Wang, X., Fan, G., Lv, Y., & Ma, J. (2023). Recent advances in Nanodrug Delivery System for Tumor Combination Treatment based on Photothermal Therapy. Advanced Therapeutics, 6(3), 2200218."
2) In lines 347, 365, 378, 413, 132, the author uses the term “laser power" and the value of 1 W/cm^2. This is not correct. It is a power density and when referring to electromagnetic radiation the term irradiance should be used (see IUPAC Recommendations 2006). The authors should correct this.
Our Response: We are sorry for this mistake.
Our Action: We corrected this mistake in the revised manuscript.
Photothermal and photodynamic therapies are carried out with very different irradiance levels. In photodynamic, the irradiance should be low to avoid heating effects. For photothermal effects, it is common to use higher irradiance levels to heat the area to be treated. The authors refer to the use of photothermal effects, but do not show that they lead directly to heating. This should be clarified by the authors.
Our response: As highlighted in the manuscript, the BPNF loaded in the bilayer generates a mild photothermal effect that increases the permeability of the bilayer and the release of doxorubicin. However, these mild hyperthermia conditions don’t significantly impact cell viability, as observed in the new phototherapy experiments using LTSL-BPNF alone.
Our action: A new set of experiments evaluating the impact of LTSL-BPNF on cell viability in different prostate and non-prostate cell lines have been added to the manuscript (new Figure 5 a,b,c).

Reviewer 3 Report
Comments and Suggestions for Authors
In this work, the authors used LTSL to load both BPNF and Dox to get LTSL-BPNF-Dox nanocomposites to increase the delivery efficiency, decrease the degradation of BPNF, and enhance the therapeutic effect of phototherapy and chemotherapy for cancer. The design profile is basically reasonable, but some data (TEM image) are blurred to analyze, some are missing or unreasonable, some design is not proper (see major part).
Major:
1. Why after loading by LTSL, the DLS size of BPNF become smaller? From 154 nm to 114 and 136 nm? I worried it’s a mixture of BPNF and LTSL-BPNF (or LTSL-BPNF-Dox), not pure LTSL-BPNF (or LTSL-BPNF-Dox).
2. In figure 2, why is there a lack of LTSL-BPNF-Dox data both for UV-Vis measurements and degradation detection?
3. Line 299-303: “The degradation of black phosphorus in water is due to the irreversible oxidation of P to phosphate groups. The initial capping reaction with DOPE and embedding the BPNF in the bilayer of LTSL greatly increases the stability of the nanomaterial.” However, there’s a lacking data to support this conclusion since the authors didn’t provide the degradation efficiency of single BPNF and did quantified comparison.
4. The release profile of Dox is weird since some points dropped even time passed, the authors should re-measure it or give some reasonable explanation. And there’s a lacking without laser groups to compare.
5. In Figure 3B, why unreleased Dox didn’t show red fluorescence? Is there any reasonable explanation or reliable reference?
6. Single phototherapy and chemotherapy are not performed both in vitro and in solution, for example, in Figure 5, the LTSL-BPNF and LTSL-Dox group should be added to see how phototherapy and chemotherapy make sense separately.
7. Why did the authors use a superoxide indicator (Dihydroethidium (DHE)) to detect only superoxide, not general ROS indicator like H2DCFDA?
Minor:
1. In vitro should be italic.
2. Line 194, “In vitro phototherapy” should be “In vitro cancer therapy” since it also includes chemotherapy.
3. What’s the yield of BPNF?
4. The TEM image looks dirty and has a large range of BPNF size, can the authors point out which part is BPNF with arrow?
5. Line 293, “red regions” should be “visible regions”, since the scope is from 220-~700 nm.
6. The lowercase logo in the image does not match the uppercase logo illustrated below the image.
Author Response
In this work, the authors used LTSL to load both BPNF and Dox to get LTSL-BPNF-Dox nanocomposites to increase the delivery efficiency, decrease the degradation of BPNF, and enhance the therapeutic effect of phototherapy and chemotherapy for cancer. The design profile is basically reasonable, but some data (TEM image) are blurred to analyze, some are missing or unreasonable, some design is not proper (see major part).
Our action: We included in Figure 1 the TEM characterisation of BPFN after exfoliation and the size analysis of the nanoflakes. The TEM image of LTSL-BPNF is provided in ESI Figure 1, and the nanoflakes loaded in the liposomes have been highlighted with arrows. The caption has been corrected for further clarification.
Major:
- Why after loading by LTSL, the DLS size of BPNF become smaller? From 154 nm to 114 and 136 nm? I worried it’s a mixture of BPNF and LTSL-BPNF (or LTSL-BPNF-Dox), not pure LTSL-BPNF (or LTSL-BPNF-Dox).
Our response: We thank the referee for highlighting this point. BPNF is generated in a top-down approach via exfoliation and centrifugation. This process can generate a polydisperse population of nanoflakes that could be difficult to separate with a mean hydrodynamic size of 154 nm. However, when we co-load the nanoflakes in the liposomes with Dox, different purification steps use a Sephadex G25 column that separates the liposomal fraction from large aggregates and the free drug that has not been encapsulated. As highlighted in ESI Figure 2, the large aggregates elute very quickly in the first fractions of the purification, followed by the liposomal fractions. This purification using a PD10 column refines the average hydrodynamic size of the formulation.
- In figure 2, why is there a lack of LTSL-BPNF-Dox data both for UV-Vis measurements and degradation detection?
Our response: We have included the LTSL-BPNF-Dox data in the general graph UV-vis graph (Figure 2a). However, we decided to study the degradation of BP in the formulation before loading Doxorubicin since the absorbance of the drug can interfere with the detection of the signal of BP, which shows a typical excitonic profile.
Our action: Modification of Figure 2a including UV-vis characterisation of LTSL-BPNF-Dox.
- Line 299-303: “The degradation of black phosphorus in water is due to the irreversible oxidation of P to phosphate groups. The initial capping reaction with DOPE and embedding the BPNF in the bilayer of LTSL greatly increases the stability of the nanomaterial.” However, there’s a lacking data to support this conclusion since the authors didn’t provide the degradation efficiency of single BPNF and did quantified comparison.
Our response: The degradation of BP in water is a well-known phenomenon described in the literature. As commented in the manuscript, the rapid oxidation of BPNF in PBS has been previously reported where the absorbance of the material in PBS decreases significantly with time, and the solution becomes colourless after 72 hours (Chem. Eng. J. 421 (2021) 127879).
- The release profile of Dox is weird since some points dropped even time passed, the authors should re-measure it or give some reasonable explanation. And there’s a lacking without laser groups to compare.
Our response: We thank the referee for highlighting the issues with the release profile. Encouraged by their suggestion, we have reassessed the experiment and improved the quality of our manuscript. The values in the figure are expressed as a percentage compared with non-irradiated samples, which have been used as (t=0).
Our Action: We have performed more release experiments and changed Figure 3a.
- In Figure 3B, why unreleased Dox didn’t show red fluorescence? Is there any reasonable explanation or reliable reference?
Our response: Due to the high encapsulation efficiency of Doxorubicin in liposomes, including LTSL, the drug forms crystals in the core of the liposomal vesicles. These aggregates keep the molecule's fluorescence quenched, and only the released Dox can be tracked using fluorescence microscopy (J. Control. Release. 328 (2020) 665–678).
Our action: The explanation and reference have been added to the manuscript (page 10): “Since the fluorescence of Dox is not quenched after its release, this enabled tracking its subcellular localisation in PC-3 cells using the RFP channel of the Muvicyte”
- Single phototherapy and chemotherapy are not performed both in vitro and in solution, for example, in Figure 5, the LTSL-BPNF and LTSL-Dox group should be added to see how phototherapy and chemotherapy make sense separately.
Our Response: Phototherapy using LTSL-BPNF and LTSL-BPNF-Dox has been assessed. Since BPNF display a synergistic PDT and PTT effect with the same laser excitation (808 nm) It would be hard to uncouple the two effects in vitro. Additionally, the aim of this work was to demonstrate how BPNF can be used as PTT agent for inducing the breaking of thermal liposomes and releasing the Doxorubicin.
- Why did the authors use a superoxide indicator (Dihydroethidium (DHE)) to detect only superoxide, not general ROS indicator like H2DCFDA?
Our Response: We thank the reviewer for this comment. DHE as well H2DCFDA are two fluorescent probes widely used for ROS detection. Nevertheless, when dealing with DHE, depending on fluorescence for detection may lead to misinterpretation. This is because these probes produce both ethidium, a byproduct of non-specific oxidation, and the O2•−-specific product 2-hydroxyethidium. The difficulty lies in the overlapping fluorescence spectra of these two products, posing a challenge in discerning the impact of non-specific oxidation compared to O2•−-dependent oxidation (if present) on the overall fluorescence signal. [Murphy, Michael P., et al. "Guidelines for measuring reactive oxygen species and oxidative damage in cells and in vivo." Nature Metabolism 4.6 (2022): 651-662.] For those reasons, we believe that DHE as well as H2DCFDA are useful to qualify ROS formation but they should not be used as quantification of ROS but to give a qualitative analysis.
Minor:
- In vitro should be italic.
Our action: Corrected
- Line 194, “In vitro phototherapy” should be “In vitro cancer therapy” since it also includes chemotherapy.
Our action: Corrected
- What’s the yield of BPNF?
Our response: The exfoliation method used in this study avoids the wastage of BP since the fraction that contains the large aggregates and is separated during centrifugation can be further sonicated and re-exfoliated. Therefore, all the starting material is converted into nanoflakes. However, as mentioned in the manuscript the encapsulation efficiency of BPNF in LTSL is 57.4%.
- The TEM image looks dirty and has a large range of BPNF size, can the authors point out which part is BPNF with arrow?
Our action: We included in Figure 1 the TEM characterisation of BPFN after exfoliation and the size analysis of the nanoflakes. The TEM image of LTSL-BPNF is provided in ESI Figure 1, and the nanoflakes loaded in the liposomes have been highlighted with arrows. The caption has been corrected for further clarification.
- Line 293, “red regions” should be “visible regions”, since the scope is from 220-~700 nm.
Corrected
- The lowercase logo in the image does not match the uppercase logo illustrated below the image.
Corrected

Round 2
Reviewer 3 Report
Comments and Suggestions for Authors
Even though the authors supplement the data and respond to the questions, there are still some issues that’s hard to explain, some small errors, and some responses are not for my questions, so further revision should be required.
Major:
1. From Fig. 1a, the flasks number seems no more than 200, how did the authors count the 200 flakes in Fig 1b?
2. There’s a big difference of DOX release profile especially for 37-degree group between old and new version, is there any possible reason? What is the meaning of “The values in the figure are expressed as a percentage compared with non-irradiated samples” in your response?
3. The author’s response “Due to the high encapsulation efficiency of Doxorubicin in liposomes, including LTSL, the drug forms crystals in the core of the liposomal vesicles. These aggregates keep the molecule's fluorescence quenched, and only the released Dox can be tracked using fluorescence microscopy (J. Control. Release. 328 (2020) 665–678).” below major question 5. However, I could find corresponding description in that paper, what I found is opposite, the DOX-loaded LTSLs shows red fluorescence in cells in Fig. 6 of that paper.
Minor:
1. The author’s response “The degradation of BP in water is a well-known phenomenon described in the literature. As commented in the manuscript, the rapid oxidation of BPNF in PBS has been previously reported where the absorbance of the material in PBS decreases significantly with time, and the solution becomes colourless after 72 hours (Chem. Eng. J. 421 (2021) 127879).” Please add this reference to your manuscript.
2. In figure 5, the caption should be removed into Experiments part, and it’s better to mark the difference between Fig. a&d, b&e, c&f in figures (like mark d, e f with laser), not only just in caption.
3. The answer for major question 7 is not answering what I asked.
Author Response
Even though the authors supplement the data and respond to the questions, there are still some issues that’s hard to explain, some small errors, and some responses are not for my questions, so further revision should be required.
We thank the Reviewer for their comments and suggestions. A detailed response follows.
Major:
- From Fig. 1a, the flasks number seems no more than 200, how did the authors count the 200 flakes in Fig 1b?
For our last revision, we took new TEM pictures of the BPNFs, counted the size of 200 flakes from different micrographs, and added the size distribution (Fig. 1b). The TEM picture in Fig. 1a is a representative image of the population.
- There’s a big difference of DOX release profile especially for 37-degree group between old and new version, is there any possible reason? What is the meaning of “The values in the figure are expressed as a percentage compared with non-irradiated samples” in your response?
We agree with the referee on this observation. We went back to analyse our lab records and realised that a less experienced operator carried out this specific experiment, and we found an error in the setup of the experiment and, therefore, the quantification of the fluorescence. Consequently, we decided to repeat the experiments and provide a new figure.
The basal fluorescence of non-irradiated liposomes is subtracted from the values obtained at different time points. 100% Dox release is assessed by dissolving the liposomes in DMSO.
- The author’s response “Due to the high encapsulation efficiency of Doxorubicin in liposomes, including LTSL, the drug forms crystals in the core of the liposomal vesicles. These aggregates keep the molecule's fluorescence quenched, and only the released Dox can be tracked using fluorescence microscopy (J. Control. Release. 328 (2020) 665–678).” below major question 5. However, I could find corresponding description in that paper, what I found is opposite, the DOX-loaded LTSLs shows red fluorescence in cells in Fig. 6 of that paper.
As reported in (J. Control. Release. 328 (2020) 665–678): "Since the fluorescence of DOX is not quenched after its release, this enabled tracking its subcellular localisation in CT26 and PC-3 cells using CLSM (Fig. 6)". The Dox molecules that can be detected from the confocal microscopy are the free one outside of the liposomes. It is worth noting that the BPNF flakes are also contributing to the Dox fluorescence quenching due to the proximity (FRET). (Biosensors 12.11 (2022): 1009; Advanced materials 29.5 (2017): 1603864.)
In the J. Control. Release. 328 (2020) 665–678 paper, we carried out the release of Dox using a water bath. Therefore, we say, “Since the fluorescence of DOX is not quenched after its release, this enabled tracking its subcellular localisation in CT26 and PC-3 cells using CLSM”
In this work, we induce the release with the laser, and only the drug released from the liposomes can be observed using the Muvicyte live imager. There’s no contradiction between both results.
We thank the reviewer for the suggested reference Advanced materials 29.5 (2017): 1603864. We included it in the manuscript (ref 25) because it also confirms that the fluorescence detected with our methodology corresponds to the free molecules outside the liposomes.
Minor:
- The author’s response “The degradation of BP in water is a well-known phenomenon described in the literature. As commented in the manuscript, the rapid oxidation of BPNF in PBS has been previously reported where the absorbance of the material in PBS decreases significantly with time, and the solution becomes colourless after 72 hours (Chem. Eng. J. 421 (2021) 127879).” Please add this reference to your manuscript.
We are sorry for this mistake. In the previous manuscript, we forgot to highlight in yellow the mentioned reference (ref. 12)
- In figure 5, the caption should be removed into Experiments part, and it’s better to mark the difference between Fig. a&d, b&e, c&f in figures (like mark d, e f with laser), not only just in caption.
We would prefer a detailed caption describing the experimental assay and data analysis. We believe that the article's readers will find it helpful to have a self-contained, self-explanatory figure and caption. We have clarified the specific treatments in each panel in the caption. We have simplified the panels of the figure as recommended by the referee.
- The answer for major question 7 is not answering what I asked.
We are sorry for this inconvenience. As reported in the literature, both Dox and BPNF can produce superoxide in cells. (Cardiovascular toxicology 21 (2021): 152-161; Journal of Hazardous Materials 454 (2023): 131502)